# A Compact Printed Monopole Antenna for WiMAX/WLAN and UWB Applications

**Zubin Chen** [1,2], **Baijun Lu** [1,2], **Yanzhou Zhu** [1,2] and **Hao Lv** [1,2,*]

[1]  Institute of Instrument Science and Electrical Engineering, Jilin University, Changchun 130026, China; czb@jlu.edu.cn (Z.C.); lubj16@mails.jlu.edu.cn (B.L.); dinggc17@mails.jlu.edu.cn (Y.Z.)
[2]  Earth Information Detection Instrument Key Laboratory of Ministry of Education, Jilin University, Changchun 130026, China
*   Correspondence: cuizl16@mails.jlu.edu.cn

**Abstract:** In this paper, a printed monopole antenna design for WiMAX/WLAN applications in cable-free self-positioning seismograph nodes is proposed. Great improvements were achieved in miniaturizing the antenna and in widening the narrow bandwidth of the high-frequency band. The antenna was fed by a microstrip gradient line and consisted of a triangle, an inverted-F shape, and an M-shaped structure, which was rotated 90° counterclockwise to form a surface-radiating patch. This structure effectively widened the operating bandwidth of the antenna. Excitation led to the generation of two impedance bands of 2.39–2.49 and 4.26–7.99 GHz for a voltage standing wave ratio of less than 2. The two impedance bandwidths were 100 MHz, i.e., 4.08% relative to the center frequency of 2.45 GHz, and 3730 MHz, i.e., 64.31% relative to the center frequency of 5.80 GHz, covering the WiMAX high-frequency band (5.25–5.85 GHz) and the WLAN band (2.4/5.2/5.8). This article describes the design details of the antenna and presents the results of both simulations and experiments that show good agreement. The proposed antenna meets the field-work requirements of cable-less seismograph nodes.

**Keywords:** antenna design; impedance bandwidth; WiMAX/WLAN band; compact printed monopole antenna

## 1. Introduction

With the rapid development of radio technology, some systems require more than a single working frequency; thus, integrating several communication standards into a single system has become a recent trend. To meet the operating frequency band requirements of IEEE 802.11 WLAN in the operating band of 2.4 GHz (2400–2484 MHz), 5.2 GHz (5150–5350 MHz), and 5.8 GHz (5725–5825 MHz), as well as the Worldwide Interoperability for Microwave Access (WiMAX) band of 2.5/3.5 GHz, the development of a multi-band antenna of 5.5 GHz (2500–2690/3400–3690/5250–5850 MHz) [1] with low cost, small size, easy manufacture, and good performance has attracted increasing attention. Recently, different architectures of multi-band antennas have been designed, including the use of microstrip feed technology to implement multi-band antennas such as an H-shaped slot antenna [2], a Y-shaped dual broadband microstrip antenna [3], a diamond-shaped slot with two U-shaped antennas [4], and an E-shaped radiation patch antenna [5], to achieve dual-band WLAN characteristics. Coplanar waveguide feed concepts have also been applied to microstrip antennas, such as the use of narrow rectangular slots with meandering asymmetric slot antennas for dual-band antennas [6], compact wide-slot antennas [7], compact asymmetric-coplanar-strip-fed tri-band meander-line antennas [8], and rectangular slot antennas [9]. Asymmetric coplanar branches have also been used for WLAN applications [10]. In addition, the use of planar inverted F antennas combined with parasitic elements

has been proposed for WLAN operation [11], an inverted L-slot triple-band antenna [12] and a CPW-fed tri-band printed antenna has been developed [13].

The antennas discussed above are suitable for WLAN communication and ground acquisition units such as the WTU-508 [14], but they are not applicable to our own developed cable-less seismograph nodes. Their practical application for cable-less seismograph nodes would contradict with their miniaturization and high-bandwidth broadband applications. Therefore, we designed an antenna that can satisfy the needs of miniaturized (<500 mm$^2$) and high-frequency bands (relative bandwidth >60%). The working scene of our self-developed seismograph is shown in Figure 1. A frequency of 2.4 GHz is used for communication between the seismograph and the AP (Access Port) relay, a frequency of 5.8 GHz is used for communication between the central control unit and the AP (Access Port) relay, and a frequency of 4.95 GHz is used for the remote seismograph information exchange with the portable hard drive.

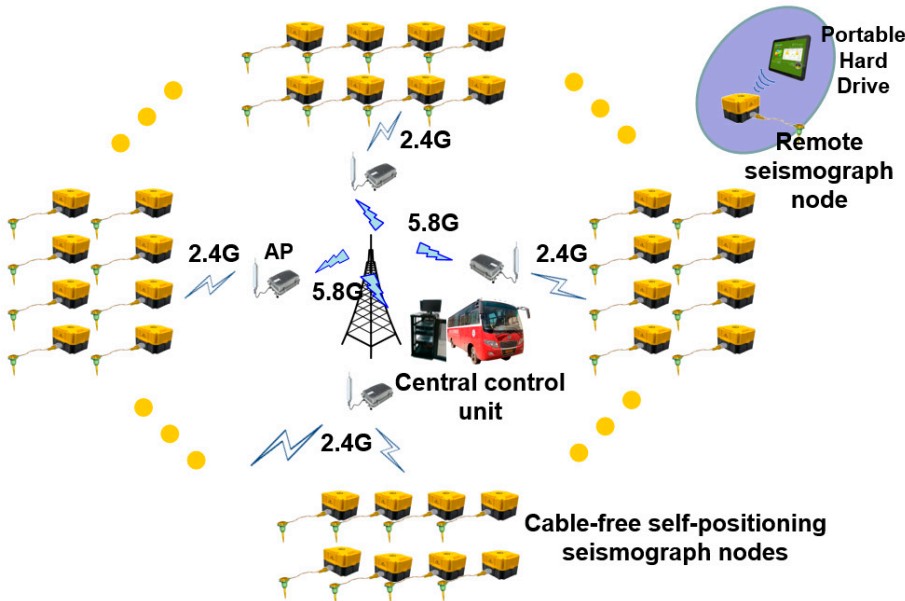

**Figure 1.** Cable-free self-positioning seismic exploration system architecture.

The ideas presented in this paper were inspired by previous structures described in the literature. The idea of a triangular structure was motivated by a previous design [15]. The meandering technique has also been previously discussed [16] and has enlightened the idea of constructing an M-type radiation patch and optimizing the structural size parameters using HFSS software. The inverted F structure was also inspired by previous work [17]. Finally, the antenna design of the present study is an extension of previous works.

This paper presents a printed monopole antenna design for WiMAX/WLAN applications for cable-free self-positioning seismograph nodes. Great improvements have been achieved in miniaturizing the antenna and in widening the bandwidth of the high-frequency band. The innovation of this design is the effective enhancement of the working bandwidth of the antenna, which was achieved by using a five-pronged feed band and a tapered impedance transformer. The feeder strip connected the triangle and two branches to form an upper-surface radiating patch. This structure was able to effectively shunt the current to generate three resonant frequencies. The M and F elements mainly served as the meandering technology to increase the current path of the antenna surface, to reduce the resonant frequency of the antenna, and to improve the bandwidth of the antenna. The M element primarily produced the 2.45-GHz band, and the F element produced the 5.80-GHz band. The ground plate consisted of rectangular patches. The miniaturization of the antenna design reduced the internal space required for the instrument. In addition, the antenna was simple in structure, easy to manufacture, and realized ultra-wideband high-frequency characteristics, thereby

improving its ability to receive signals. This antenna proved to be suitable for the WLAN/WiMAX (5.15–5.35 GHz) high-frequency band. This article presents the detailed geometric configuration of the antenna and discusses some of its parameters, including the size of the radiation patch and ground patch. A comparison of this antenna with several antennas proposed in recent years for WLAN/WiMAX applications is presented in Table 1. Compared with the other antennas, it is apparent that the proposed antenna is very small and provides a very wide bandwidth at high frequencies, completely covering the WLAN frequency band. Although the relative bandwidth of an antenna in the high-frequency band outlined in a previous study [3] does not differ greatly from that of the design proposed in this paper, the current design has the advantages of a simple and compact structure, small volume, and easy manufacture for small-volume high-frequency antenna applications. The second part of the article covers the design of the antenna, the third part presents an analysis of the current distribution and parameters, and the fourth part presents the simulation and test results. Concluding remarks are provided in the fifth part.

**Table 1.** Comparison of microstrip multi-band antennas.

| References | Type | Size (mm$^2$) | Total Area (mm$^2$) | Bandwidth (GHz) | Relative Bandwidth |
|---|---|---|---|---|---|
| [2] | Tri-band | 60 × 60 | 3600 | 1.55–1.57, 2.39–2.69, 4.97–5.93 | 1.27%, 12.00%, 16.55% |
| [3] | Dual-band | 35 × 24 | 840 | 2.26–2.67, 3.00–6.78 | 16.40%, 65.17% |
| [4] | Dual-band | 40 × 40 | 1600 | 3.15–3.70, 5.05–5.97 | 15.70%, 15.86% |
| [5] | Dual-band | 40 × 30 | 1200 | 2.39–2.51, 5.00–6.10 | 4.80%, 18.96% |
| [6] | Dual-band | 34 × 30 | 1020 | 2.30–2.50, 2.90–15.00 | #, #, # |
| [7] | Tri-band | 40 × 40 | 1600 | 2.28–2.58, 3.38–3.66, 5.07–5.86 | 12.60%, 8.00%, 14.50% |
| [8] | Tri-band | 35 × 15 | 525 | 1.48–1.63, 2.25–2.48, 4.22–6.00 | 9.55%, 9.38%, 30.69% |
| [9] | Dual-band | 75 × 75 | 5625 | 2.40–2.48, 5.15–5.95 | 10.60%, 33.8% |
| Proposed work | Dual-band | 30 × 17 | 510 | 2.39–2.49, 4.27–7.96 | 4.08%, 64.31% |

**Notation:** # represents information not mentioned in the reference.

## 2. Antenna Design

The geometric configuration of the proposed antenna is shown in Figure 2. The antenna consisted of a triangle, an inverted F, and an M, which was rotated 90° counterclockwise to form a surface-radiating patch. This antenna was fabricated on an FR4_epoxy substrate with a relative permittivity of 4.4, a tangent of 0.02, and a height of 1.6 mm. The size of the antenna substrate was 0.25 λ × 0.14 λ × 1.6 mm (30 × 17 mm, λ is the free wavelength in a 2.45-GHz space). This structure was fed by a 50-Ω microstrip transmission line with length $L_f$ and width $W_f$. The antenna ground plate was composed of a rectangular patch with length W and width $L_g$. The two branches of the antenna increased the flow path of the surface current. In this antenna geometry model, the resonant path lengths $L_{11}$ and $L_{22}$ were approximately one-quarter of the free wavelength in air at the corresponding frequency. The proposed antenna covered the WLAN (2.4/5.2/5.8) and WiMAX high-frequency (5.25–5.85 GHz) bands.

$L_{11}$ and $L_{22}$ can be described by Equations (1) and (2):

$$L_{11} = L_2 + L_7 + L_4 + L_6 + 2W_6 + W_7 + 3a \tag{1}$$

$$L_{22} = L_2 + L_3 + L_4 \tag{2}$$

From the design of the microstrip patch antenna, the theoretical dimensions were calculated as follows [18]:

The width $w$ of the patch is:

$$w = \frac{c}{2f_0}\left(\frac{\varepsilon_r + 1}{2}\right)^{-\frac{1}{2}} \tag{3}$$

where $c$ represents the speed of light, $f_0$ is the center frequency of the antenna, and $\varepsilon_r$ is the relative permittivity of the dielectric substrate. The relative effective dielectric constant of the dielectric substrate $\varepsilon_{re}$ is:

$$\varepsilon_{re} = \frac{\varepsilon_r + 1}{2} + \frac{\varepsilon_r - 1}{2}\left(1 + \frac{12h}{w}\right)^{-\frac{1}{2}}. \tag{4}$$

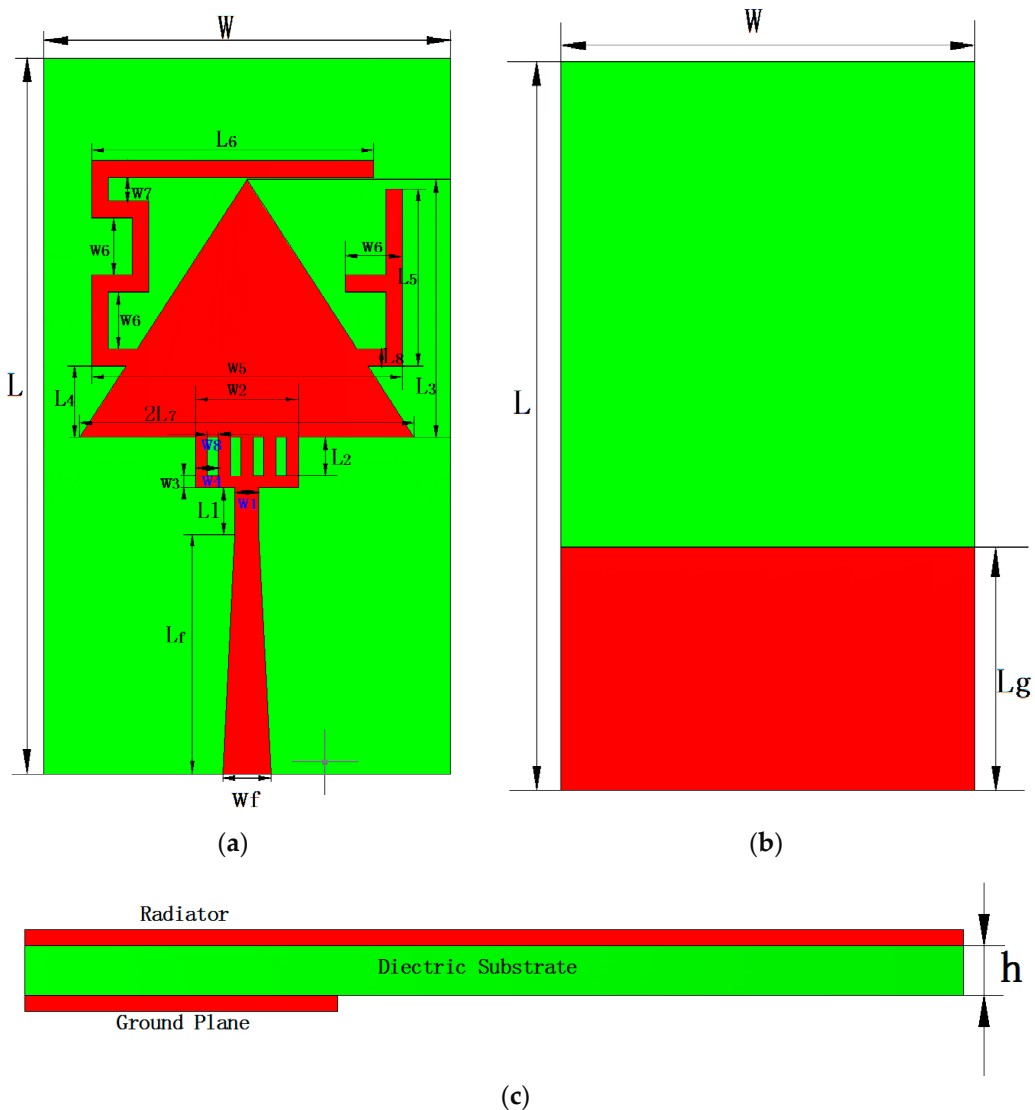

**Figure 2.** Geometrical configuration of the proposed antenna: (**a**) top view, (**b**) bottom view, and (**c**) side view.

The equivalent extension length $\Delta L$ caused by the fringe field of the antenna is:

$$\Delta L = 0.412h\frac{(\varepsilon_{re} + 0.3)\left(\frac{w}{h} + 0.264\right)}{(\varepsilon_{re} - 0.258)\left(\frac{w}{h} + 0.8\right)} \tag{5}$$

where $h$ represents the thickness of the dielectric substrate.

From this equation, it is possible to calculate the actual length L of the rectangular patch:

$$L = \frac{c}{2f_0}\frac{1}{\sqrt{\varepsilon_{re}}} - 2\Delta L \tag{6}$$

The fundamental dimensions of the antenna are based on the above four equations. First, we designed a structure of a three-band antenna. The specific size was derived from High-Frequency Electromagnetic Simulation Software (HFSS). Then we optimized the parameters calculated by HFSS to find the parameters which best meet the design requirements. The specific size parameters for the antenna are listed in Table 2.

**Table 2.** Design parameters of the proposed antenna.

| Parameters | Unit (mm) | Parameters | Unit (mm) | Parameters | Unit (mm) |
|---|---|---|---|---|---|
| W | 17.00 | $W_7$ | 1.00 | $L_5$ | 7.40 |
| $W_f$ | 2.00 | $W_8$ | 0.45 | $L_6$ | 11.80 |
| $W_1$ | 1.00 | L | 30.00 | $L_7$ | 7.00 |
| $W_2$ | 4.30 | $L_f$ | 10.00 | $L_8$ | 0.70 |
| $W_3$ | 0.50 | $L_1$ | 2.00 | $L_g$ | 10.00 |
| $W_4$ | 0.95 | $L_2$ | 1.60 | $h$ | 1.60 |
| $W_5$ | 13.00 | $L_3$ | 10.80 | | |
| $W_6$ | 2.40 | $L_4$ | 3.00 | | |

Figure 3a,b present photographs of a manufacturing prototype of the proposed antenna. A layer of tin was attached to the surface to prevent the oxidation of the radiation surface of the antenna. The design steps of the proposed antenna employed in HFSS are illustrated in Figure 3c. Figure 4 shows the return loss versus frequency diagram for the three cases. For Antenna 1 (Ant1), there was only a gradual microstrip line and a triangular radiating patch, and a single frequency band appeared. The bandwidth at the resonance frequency of 5.68 GHz was in the range of 4.59–7.88 GHz. For Antenna 2 (Ant2), dual-frequency characteristics were exhibited with respect to the Ant1 antenna, and resonance frequency appeared at 2.60 GHz. The matching of these antennas was not very good and did not meet the requirement of 2.40 GHz for Bluetooth. Finally, the antenna structure of Antenna 3 (Ant3) offered improvements. This antenna covered two frequency bands of 2.39–2.49 and 4.26–7.99 GHz, and the bandwidth of the high-frequency band was clearly wider than that of Ant1. This antenna design meets the requirements of the WLAN and WiMAX (5.25–5.85 GHz) bands. All the optimized dimensions are the same as those listed in Table 2.

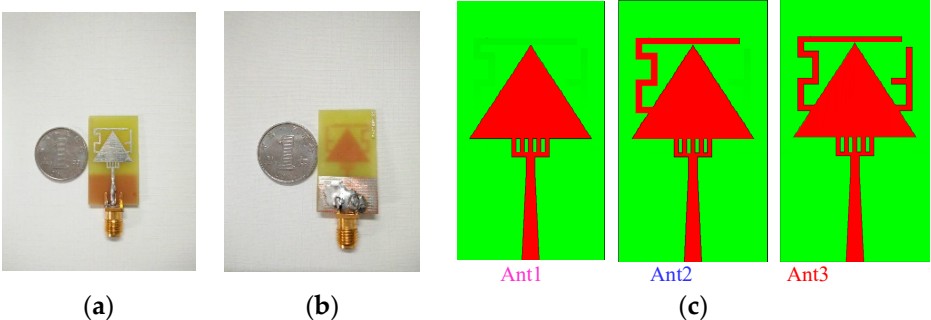

(**a**)        (**b**)        (**c**)

**Figure 3.** Fabricated photograph and design steps for the proposed antenna. (**a**) Top view, (**b**) bottom view, and (**c**) design steps for the proposed antenna used in HFSS software.

Figure 5 presents the simulation result of the voltage standing wave ratio (VSWR) for the proposed antenna as a function of frequency. In wireless communication, this parameter indicates the matching degree of the antenna and feeder. When VSWR equals 1, the impedance is precisely matched to achieve the maximum power transmission. The larger the standing wave ratio, the higher the reflected power and the lower the transmission efficiency. The VSWR of the antenna was less than 1.5 at 2.45 and 5.80 GHz. Figure 6 presents the input impedance curve of the antenna. The input impedance at 2.45 and 5.80 GHz was 51.0049$-$8.1573 j and 48.8302$-$0.3752 j, respectively. These results indicate that the input impedance is almost 50 $\Omega$ at the resonant frequency and that the input impedance of the

antenna is very stable in the 2–10-GHz frequency bands and very close to 50 Ω. The main purpose of
the antenna impedance is to achieve a matching between the antenna and the feeder. If the transmit
antenna matches the feed line, the input impedance of the antenna should be equal to the characteristic
impedance of the feed line. A Smith chart of the antenna is presented in Figure 7, showing information
on the antenna impedance matching, the standing wave ratio, and the normalized impedance. Ang and
Mag represent the phase and amplitude at the resonance points in polar coordinates, respectively.
Rx represents the input impedance of the antenna. It can be seen that the antenna has a good normalized
impedance at the frequencies studied. Table 3 lists the specific values of the Smith chart in Figure 7.

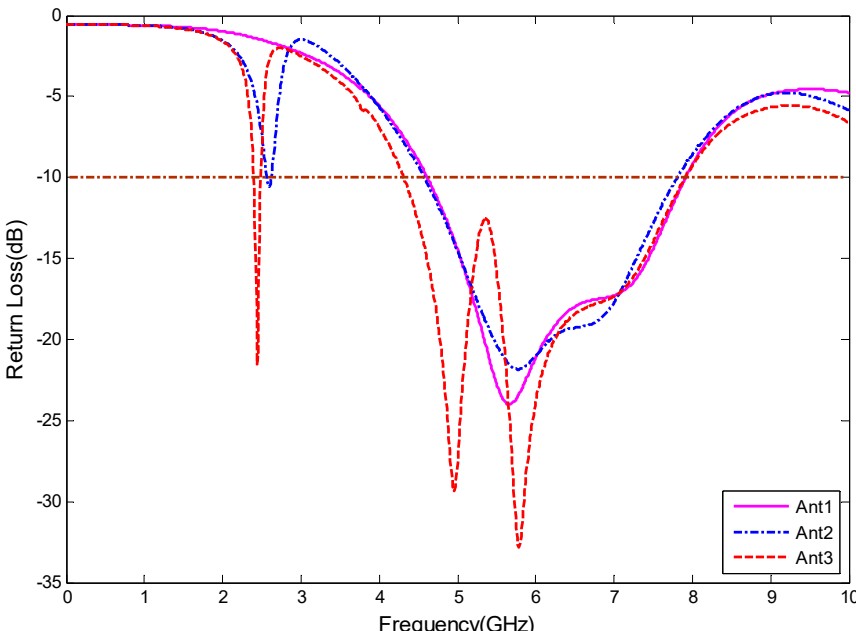

**Figure 4.** Simulated return loss versus frequency graphs for Antenna 1 (Ant1), Antenna 2 (Ant2), and
Antenna 3 (Ant3) (Ant3 being the proposed antenna).

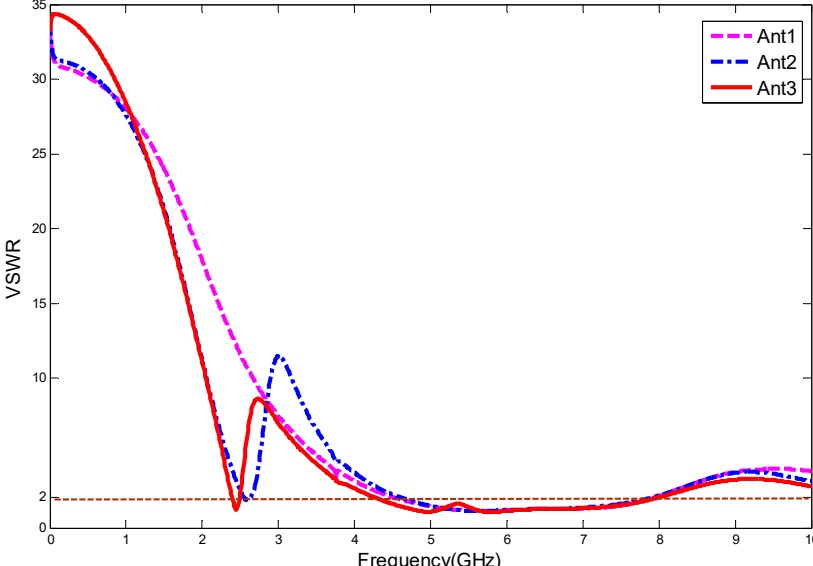

**Figure 5.** Simulated VSWR (voltage standing wave ratio) versus frequency for Ant1, Ant2, and Ant3
(the proposed antenna).

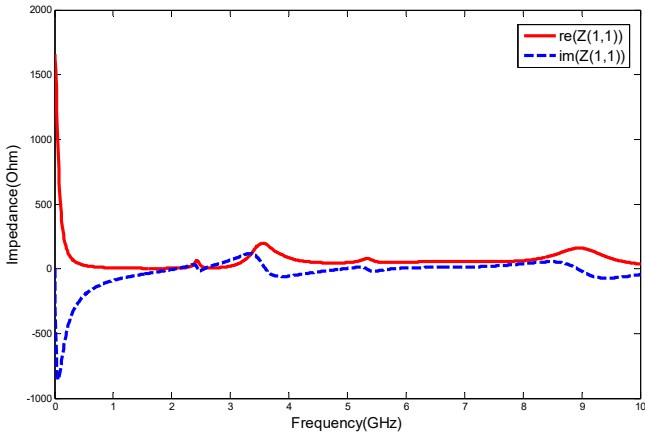

**Figure 6.** Input impedance curve of the proposed antenna.

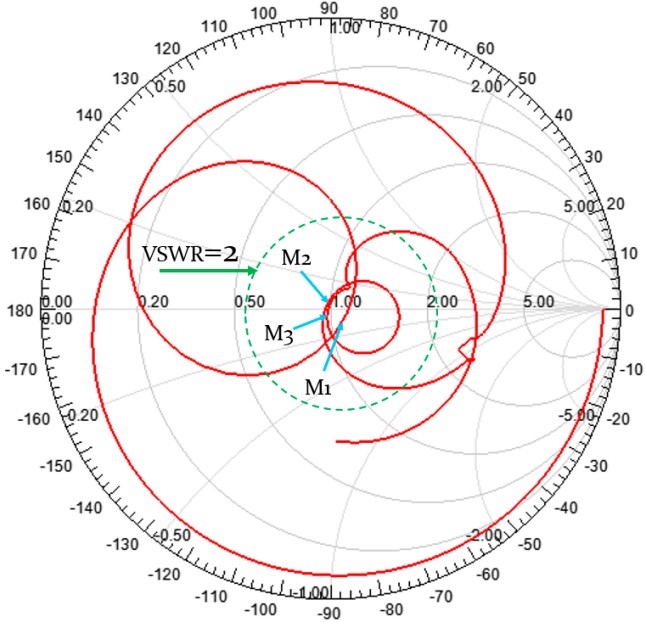

**Figure 7.** Smith chart of simulated antenna matched to 50 Ω.

**Table 3.** Smith chart value of the proposed antenna.

| Name | Freq | Ang | Mag | Rx | VSWR |
|------|------|-----|-----|-----|------|
| M1 | 2.45 | −49.9679 | 0.0862 | 1.1072 − 0.1473i | 1.1887 |
| M2 | 4.95 | 156.7160 | 0.0344 | 0.9385 + 0.0255i | 1.0712 |
| M3 | 5.80 | 155.2614 | 0.0236 | 0.9578 + 0.0189i | 1.0484 |

## 3. Current Distribution and Parameter Analysis

Figure 8a–c present the simulation results of the surface current distribution of the antenna at 2.45, 4.95, and 5.80 GHz, respectively. At 2.45 GHz, a strong current distribution was observed at the tip of the triangular radiating patch, the right end of the M structure, and the bottom of the inverted F structure, which mainly affected the low-frequency resonant frequency of the antenna. The surface current distribution at 4.95 GHz revealed a strong current at the bottom of the feeder and at the bottom of the triangle. At 5.80 GHz, the current was distributed over the feeder as well as at several other locations. This figure clearly demonstrates the main influencing factors of the resonant frequency of the corresponding points. We were therefore able to adjust the corresponding frequency depending on the main distribution of the current to change the resonant frequency.

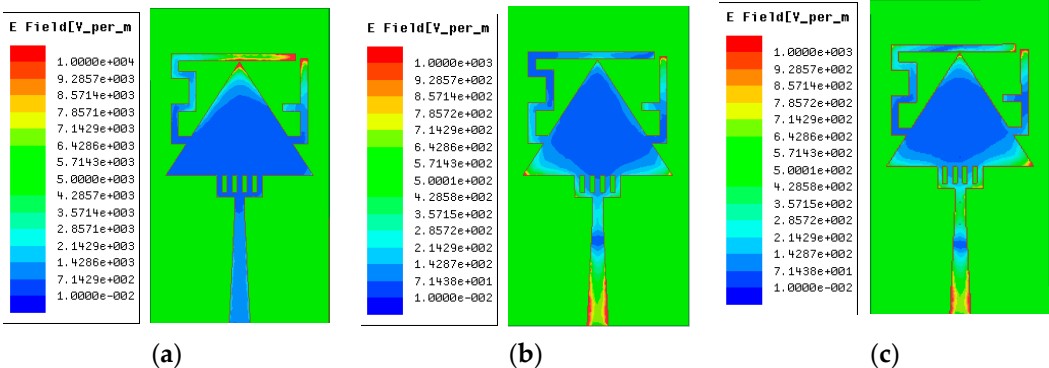

(**a**)                    (**b**)                    (**c**)

**Figure 8.** Surface current distribution (V/m) of the proposed antenna at (**a**) 2.45 GHz, (**b**) 4.95 GHz, and (**c**) 5.80 GHz.

Figures 9–12 depict the simulated reflection loss effect for different values of $L_6$, $L_5$, $L_7$, and $L_g$. The effect of the length of one arm of the upper-surface radiating patch on the return loss parameter of the antenna is shown in Figure 9. The length of $L_6$ had a significant effect on the 4.95-GHz resonant frequency of the antenna. As the size of $L_6$ increased from 11.6 to 12 mm, a slight offset occurred at all three resonance points and the matching of the antenna deteriorated. Therefore, the optimal solution was achieved when $L_6$ was 11.8 mm. The results of the analysis of the length of the inverted F structure are presented in Figure 10. The effect of this parameter on the three frequencies was large, and when $L_5$ increased from 7.3 mm to 7.5 mm, the resonant frequency first increased and then decreased. The match to 5.80 GHz gradually deteriorated, and the matching characteristic of 4.95 GHz was exactly opposite that at 5.80 GHz. Finally, the optimal value of $L_5$ was 7.4 mm to ensure function in the Bluetooth band of 2.4 GHz. Figure 11 presents the length analysis results for the bottom edge parameters of the triangle. With increasing $L_7$, the three resonant frequency points all show a slight offset. Therefore, this parameter can be used to fine-tune the resonant frequency of the antenna. The matching effect in the high-frequency and low-frequency bands did not change greatly. The selection of 7 mm for $L_7$ satisfied the requirements of the antenna better. Figure 12 shows that the width $L_g$ of the antenna ground plane mainly affected the matching problem of the antenna. It had a great influence on the medium and high frequencies. The low resonant frequency of the antenna also had a substantial effect, and an optimal value for $L_g$ of 10 mm was selected.

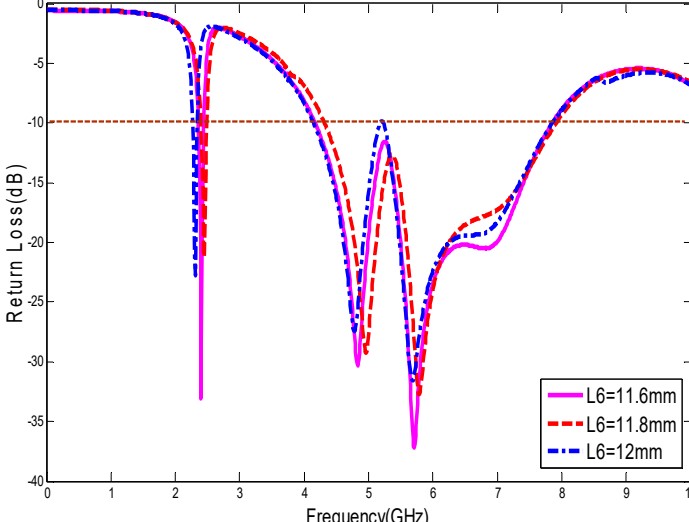

**Figure 9.** Simulated return loss versus frequency for the proposed antenna for various values of $L_6$; other parameters are the same as those listed in Table 2.

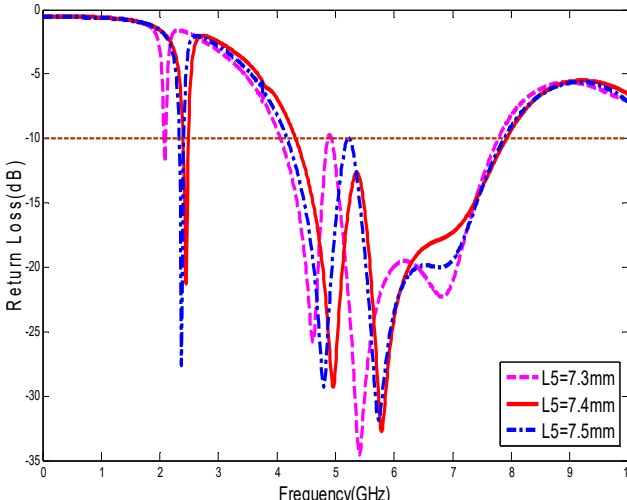

**Figure 10.** Simulated return loss versus frequency for the proposed antenna for various values of L₅; other parameters are the same as those listed in Table 2.

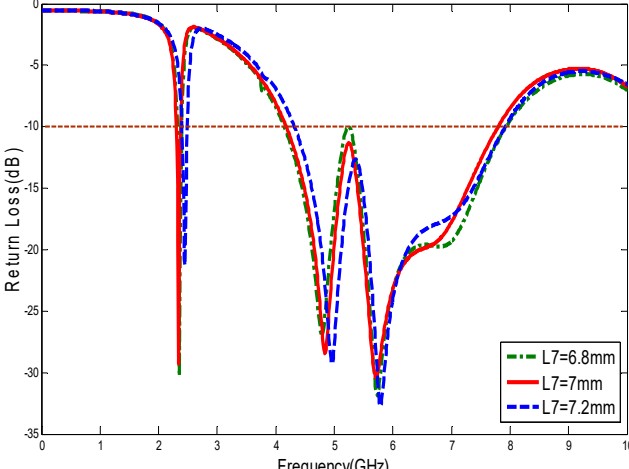

**Figure 11.** Simulated return loss versus frequency for the proposed antenna for various values of L₇; other parameters are the same as those listed in Table 2.

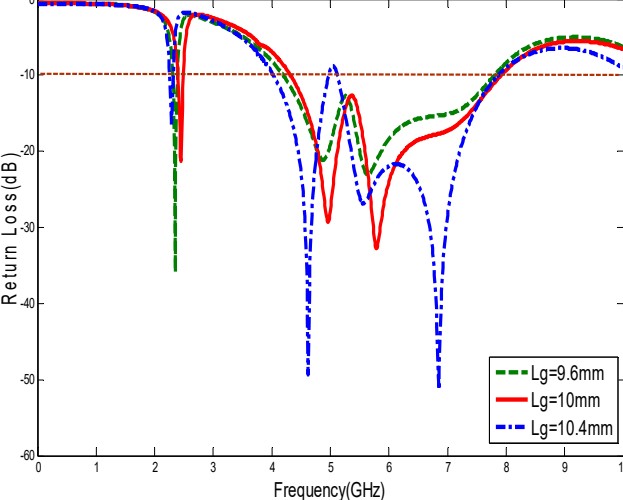

**Figure 12.** Simulated return loss versus frequency for the proposed antenna for various values of L_g; other parameters are the same as those listed in Table 2.

## 4. Experimental Results and Discussion

The Agilent E5071C vector network analyzer was used to test the return loss, VSWR, and forward transmission parameters S21 of the antenna. Figure 13 presents the results of the simulation and experiment for the return loss of the antenna. The simulation results are in good agreement with the experimental results. The errors may originate from SMA (Sub-Miniature-A) head connection welding, substrate loss, the antenna size being too small, the inaccuracy of the processing dimensions, or the test environment. The differences may also be associated with the relative permittivity used in the simulation and loss tangent uncertainty caused by error. Figure 14 presents the VSWR curve for the antenna simulation and the actual measurement. The VSWR was less than 2 in the range of 4.20–8.20 GHz near 2.34 GHz. The simulation and experimental results were generally consistent. Figure 15 presents the results for the forward transmission parameter S21 between the two antennas. It can be observed from the figure that the transmission performance is good at the frequencies discussed.

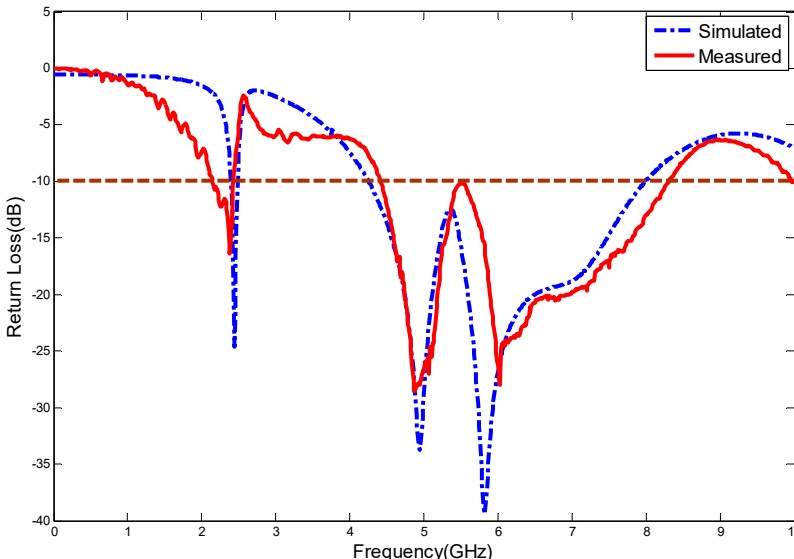

**Figure 13.** Simulated and measured return loss of the proposed antenna.

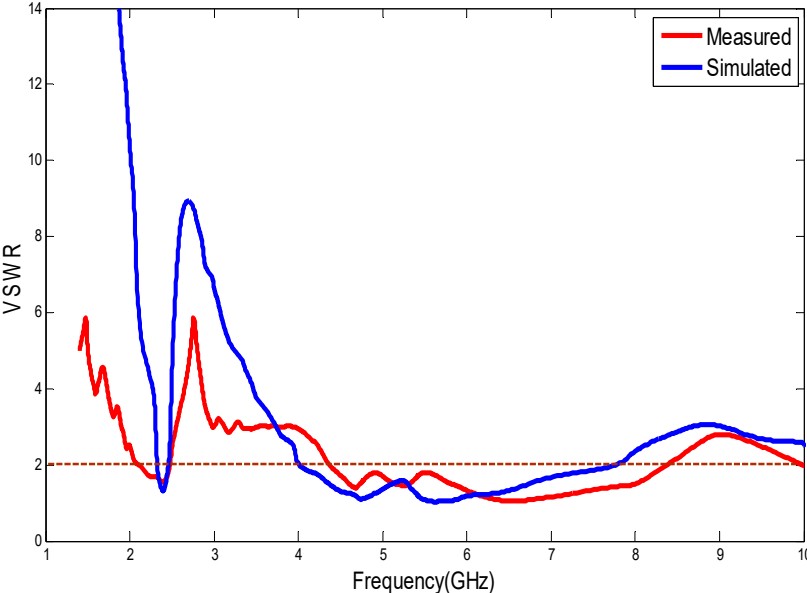

**Figure 14.** Simulated and measured VSWR of the proposed antenna.

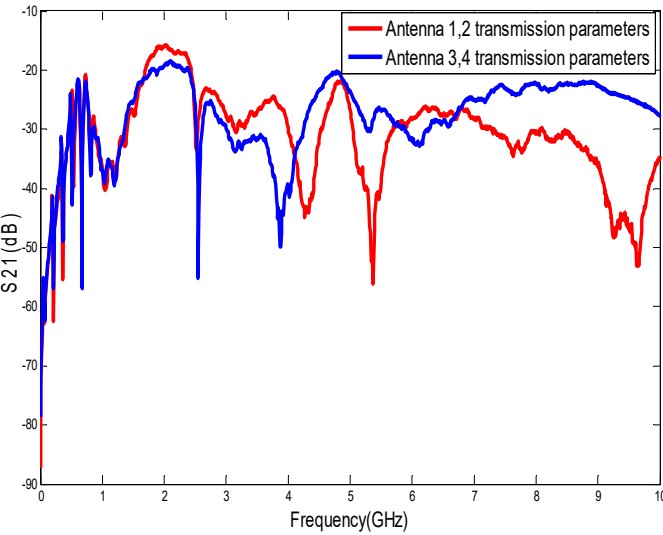

**Figure 15.** Transmission parameter curve of the proposed antenna.

The gain curve of this antenna is presented in Figure 16, and the gains at 2.45, 4.95, and 5.80 GHz were approximately −3.57, 1.61, and 0.95 dB, respectively. The radiation efficiencies were approximately 47.27%, 95.72%, and 95.58%, respectively. This antenna showed excellent radiation efficiency at high frequencies. Table 4 also summarizes the gain and efficiency of the proposed antenna. Figure 17 shows a photograph of the test environment of the antenna forward transmission parameter S21.

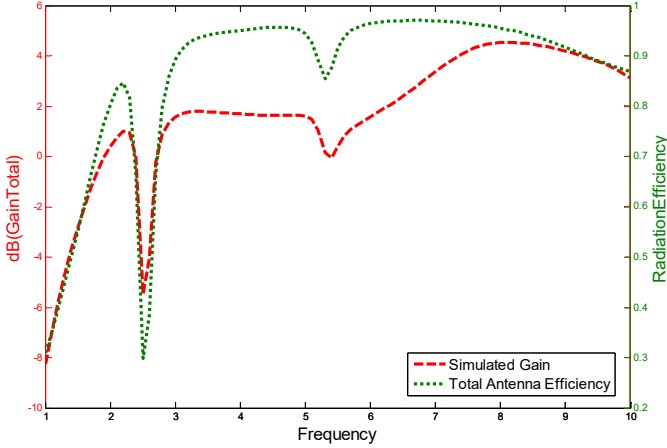

**Figure 16.** Simulated gain and total antenna efficiency of the proposed antenna.

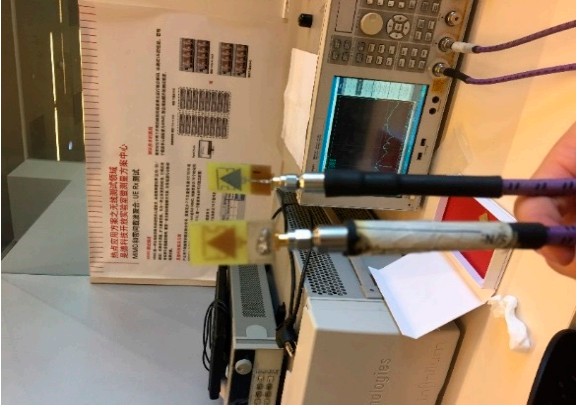

**Figure 17.** S21 test environment of the proposed antenna.

**Table 4.** Gain and efficiency of the proposed antenna.

| Frequency | 2.45 (GHz) | 4.95 (GHz) | 5.80 (GHz) |
|---|---|---|---|
| Gain (dB) | −3.57 | 1.61 | 0.95 |
| Efficiency | 47.27% | 95.72% | 95.58% |

Table 5 compares the performance of the antenna proposed in this paper with existing antennas. The gain at 5.8 GHz of the first reference antenna design [5] and the efficiency at 2.45 GHz of the second reference antenna design [19] are slightly better than the antenna of the present study. Other parameters of the design proposed in this paper are superior to the three reference antennas. The relative bandwidth of the antenna proposed in this paper reached 64.31% in the high-frequency band, which is far superior to that of the reference antennas [5,19,20]. This paper focuses on miniaturization [21] and high-frequency bandwidth [22] and the proposed antenna has great advantages in the above aspects. In the future, this antenna can be applied in the field of high-frequency broadband antennas.

**Table 5.** Comparison between the proposed and some existing antennas.

|  | Frequency | 2.45(GHz) | 4.95(GHz) | 5.80(GHz) | Relative Bandwidth |
|---|---|---|---|---|---|
| [5] | Gain (dB) | −0.90 | 1.58 | 2.20 | 4.80% |
|  | Efficiency | 52.00% | 75.00% | 89.00% | and 18.96% |
| [19] | Gain (dB) | 3.08 | −0.5 | 0.4 | 23.3% |
|  | Efficiency | 91.07% | 22% | 45% | and 14.03% |
| [20] | Gain (dB) | −0.70 | 1.60 | 1.65 | # |
|  | Efficiency | # | # | # | # |
| Proposed antenna | Gain (dB) | −3.57 | 1.61 | 0.95 | 4.08% |
|  | Efficiency | 47.27% | 95.72% | 95.58% | and 64.31% |

**Notation:** # represents information not mentioned in the reference.

Figure 18 presents a radiation pattern of the proposed antenna and two-dimensional far-field radiation patterns of the E-plane and H-plane at different frequencies. As can be seen from the figure, the antenna proposed in this paper has a good radiation characteristic. In the E-plane, the radiation direction is oriented toward 0° and 180°, presenting a dipole-like radiation pattern. The E-plane radiation pattern of the antenna is in the shape of a figure eight, which indicates maximum radiation in the directions of 0° and 180°. In the H-plane, except for a singularity at 4.95 GHz, the other frequencies are all close to omni-directional.

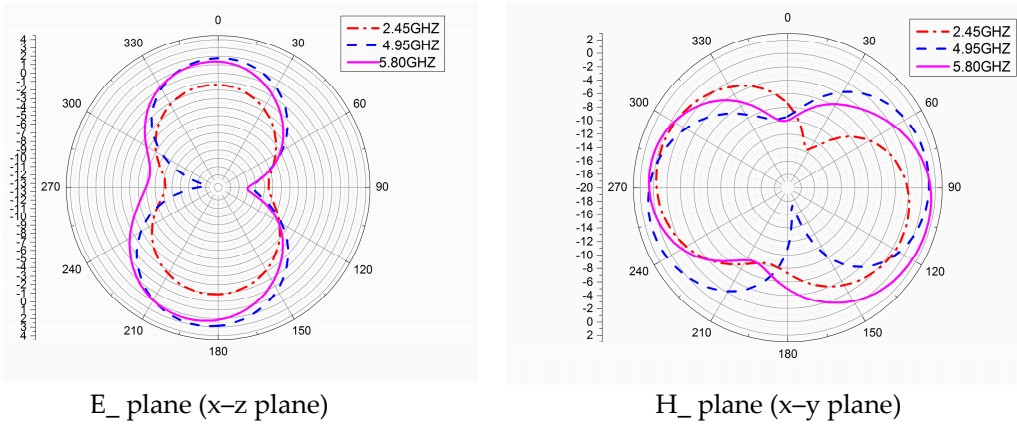

E_ plane (x–z plane)          H_ plane (x–y plane)

**Figure 18.** Radiation patterns of the proposed microstrip UWB antenna.

## 5. Conclusions

In this paper, a design for a printed monopole antenna suitable for cable-less seismograph nodes is proposed. This antenna can work in the WiMAX (5.25–5.85 GHz) and WLAN (2.4, 5.2, and 5.8 GHz) bands. The proposed antenna satisfies the needs for miniaturization and has a relative bandwidth of 64% in the high-frequency band. The prototype with a compact overall size of 17 mm × 30 mm × 1.6 mm achieved a measured bandwidth of 270 MHz (2.23–2.50 GHz) and 3.88 GHz (4.30–8.18 GHz) at lower and upper bands, respectively. The simulation and experimental results generally agree well with each other. The test results indicate a high-frequency bandwidth of 4.30–8.18 GHz for the proposed antenna. Therefore, the proposed simple antenna may be suitable for next-generation UWB systems. Of course, we can also improve the gain of the antenna to increase the transmission distance. Thereby, it would connect more cable-less seismograph nodes under the same system. Thus, we could reduce the number of AP (Access Port) relays.

**Author Contributions:** The data collection is completed Z.C. The research design is completed by B.L. The document retrieval and icon production are completed by Y.Z. The software part is completed by H.L.

**Funding:** The research is sponsored by the Natural Science Foundation of China (No.41404097, 41074074), Science and technology development project of Jilin Province (No.20150520071JH, 20160204065GX, SXGJSF2017-5), China Postdoctoral Science Foundation (No.2015M571366).

**Conflicts of Interest:** The authors declare no conflict of interest.

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
