# Peer review of "A Compact Printed Monopole Antenna for WiMAX/WLAN and UWB Applications"

_futureinternet, doi:10.3390/fi10120122_

Round 1

Reviewer 1 Report

The present work presents interest on the area of antenna development. The authors must revise their English. The phrases are large and the concept of the idea get loosed. Please make the phrases shortly.

The present type of antenas are very common in research. Also the authors try to improve the quality of these antenas. The work needs to be improved with more discutions. In the page 10 lines 208, The gain…. The authors describe the comparative of their results with two other developments. There I understud that both referencies obtained better results that the proposed design. They mention that the bandwich increase over 64,31% respect to other antenas (miss references, with which antenas is the comparation). 

Author Response

The present work presents interest on the area of antenna development. The authors must revise their English. The phrases are large and the concept of the idea get loosed. Please make the phrases shortly.

The present type of antennas are very common in research. Also the authors try to improve the quality of these antennas. The work needs to be improved with more discutions. In the page 10 lines 208, The gain…The authors describe the comparative of their results with two other developments. There I understud that both referencies obtained better results that the proposed design. They mention that the bandwich increase over 64.31% respect to the antennas (miss references, with which antennas is the comparation).

English has been modified.

Although the antenna discussed above is suitable for WLAN communication, its practical application for cableless seismograph nodes cannot meet its miniaturization and high-bandwidth broadband applications. Therefore, the antenna designed in this paper is proposed. And processed the antenna to test its parameters. The design proposed in this paper satisfies the needs of miniaturized (<550mm2) and high frequency bands (relative bandwidth >60%). In the page 10 lines 208, a simple comparison of gain and efficiency, although individual parameters are not as good as 5.80 GHz for [5], 2.45 GHz for [18], and 5.80 GHz for [19]. The efficiency and gain at other points are superior to these three antennas, and this paper has great advantages in miniaturization and high-frequency bandwidth of the main research.

Reviewer 2 Report

The work is very interesting for a dynamic spectrum access. 

Author Response

Response: Thank you very much.

Reviewer 3 Report

Introduction

1.      Missing citations for lines 26-27; lines 28-29

2.      Lines 50-51 – what is the novel contribution and why?

3.      Line 56 – what are the 3 resonant frequencies – what is so special about this?

4.      Lines 59-60 – citation

Missing Related Work

There is a lot of research on monopole, WLAN and UWB. Review them, identify gaps and  describe how your research address them.

Antenna Design and Subsequent Sections

1.      Line 78 Fig. 1 – too far away. The diagram is not legible.

2.      Line 86 – write L11 and L22 as equations

3.      Equation 1 – what is Er?

4.      Equation 2 – what is w? If it is width then make sure the case is consistent with equation 1

5.      Equation 3 – what is h?

6.      Equation 4 what is delta L?

7.      Tables 1 and   4 - #? Use separators for easy reading. No of decimal places should be consistent

8.      Table 2 – aggregate similar parameters together. Use vertical separators. No of decimal points should be consistent

9.      Line 114 – figure number?

10.  Line 110 – how to assess the improvement made by Ant3 and why?

11.  Line 114 – Figures a, b, c – what do they represent? Provide some explanation

12.  Figure 2 to 16 – provide keys. Elaborate on explanation and provide interpretation of whatever that is depicted. Provide the reasons for whatever that is presented in the diagrams/graphs

13.  Figure 5, Separate diagram from the table

14.  Lines 123-132 – discussion could be more elaborate, and critical.  Describe the graphs, interpret them followed by providing possible reasons for the depicted results

15.  Line 158  - Figures 7-10 (repeat 14)

16.  Conclusion – lacks rigour

Author Response

Missing citations for lines 26-27; lines 28-29.

Response: The citations for lines 26-27; lines 28-29 has been added. Thank you very much.

Lines 50-51 – what is the novel contribution and why?

Response: Since the existing antenna cannot meet the working requirements of our cableless seismograph nodes, mainly because of its large size and narrow frequency band, the design of this paper is proposed. The innovation of this design is the effective enhancement of the working bandwidth of the antenna achieved by using the five-pronged feed band and tapered impedance transformer. Great improvements have been achieved in the miniaturization of the antenna and in widening the bandwidth of the high-frequency band. Thank you very much. 

Line 56 – what are the 3 resonant frequencies – what is so special about this?

Response: The 3 resonant frequencies are 2.45GHZ、4.95GHZ and 5.80GHZ. The “M” element primarily produces the 2.45-GHz band, and the “F” element produces the 5.80-GHz band. The triangular radiation patch produces a 4.95 GHz band. Multi-band operation is achieved with a new structure of current zigzag technology. The 2.4GHZ and 5.80GHZ bands are primarily used for data transmission in cableless seismograph nodes. (2.4GHZ is used for communication between seismograph nodes and AP relay, 5.8GHZ is used for communication between central control unit and AP relay, and 4.95GHZ is used for information exchange between seismograph nodes and Portable Hard Driver.) That’s all. Thank you.

Lines 59-60 – citation

Response: This structure is designed independently by myself. The main citations section has been given in lines 44-49. The “M” element primarily produces the 2.45-GHz band, and the “F” element produces the 5.80-GHz band. This conclusion is derived from the analysis of the Return Loss curves of the antenna design steps Ant1, Ant2 and Ant3 and the analysis of the antenna surface current distribution. That’s all. Thank you.

Antenna Design and Subsequent Sections

Line 78 Fig. 1 – too far away. The diagram is not legible.

Response: Figure 1 has been enlarged. That’s all. Thank you.

Line 86 – write L11 and L22 as equations.

Response: Line 86 L11 and L22 as equations have been described as (1) and (2). That’s all. Thank you.

Equation 1 – what is Er?

Response: ε_r is the relative permittivity of the dielectric substrate. ε_re is Relative effective dielectric constant of dielectric substrate. That’s all. Thank you.

Equation 2 – what is w? If it is width then make sure the case is consistent with equation 1.

Response: W is the width of the patch. W is consistent with equation 1. That’s all. Thank you.

Equation 3 – what is h?

Response: h represents thickness of the dielectric substrate. That’s all. Thank you.

Equation 4 what is delta L?

Response: ∆L is the equivalent extension length caused by the fringe field of the antenna. That’s all. Thank you.

Tables 1 and   4 - #? Use separators for easy reading. No of decimal places should be consistent.

Response: It is now easy to read. Tables 1 and   4, the number of decimal places is consistent. That’s all. Thank you.

  Table 2 – aggregate similar parameters together. Use vertical separators. No of decimal points should be consistent.

Response: Similar parameters have been put together. Use vertical separators. The number of decimal places is consistent. That’s all. Thank you.

Line 114 – figure number?

Response: Line 114 – figure number is added. That’s all. Thank you.

Line 110 – how to assess the improvement made by Ant3 and why?

Response: Ant3 is to implement WLAN (2.45GHZ) communication for the function of this design. This application was not implemented at 2.45 GHz compared to Ant2. And Ant3 also makes the matching characteristics of the antenna better and appropriately increases the high frequency bandwidth of the antenna. Ant3 is better able to achieve the target function. That’s all. Thank you.

Line 114 – Figures a, b, c – what do they represent? Provide some explanation.

Response: Line 114 – Figures a, b represent the upper and lower surfaces of the antenna prototype, respectively. Figure c represent design steps for the proposed antenna used in HFSS software. That’s all. Thank you.

Figure 2 to 16 – provide keys. Elaborate on explanation and provide interpretation of whatever that is depicted. Provide the reasons for whatever that is presented in the diagrams/graphs.

Response: All the diagrams/graphs have been explained in the paper. These diagrams/graphs show the various parameters that must be considered for the antenna design. That’s all. Thank you.

Figure 5, Separate diagram from the table.

Response: Diagram and the table have been separated. That’s all. Thank you.

Lines 123-132 – discussion could be more elaborate, and critical.  Describe the graphs, interpret them followed by providing possible reasons for the depicted results.

Response: Lines 123-132 describes a re-detailed description of the chart. Of course, there will be no perfect situation, and electromagnetic waves will have energy loss during the propagation process. The antenna being too small, the accuracy of the processing dimensions or the test environment may also cause errors. These explain the deviation between the parameters in the text and the ideal case. That’s all. Thank you.

Line 158  - Figures 7-10 (repeat 14)

Response:  Figures 7-10(Figures 9-12 now) have given detailed explanations in the text.

 First, we can observe the main part affecting the antenna frequency by analyzing the current intensity of the antenna surface. For example, at 2.45 GHz, there is a strong current distribution on the right side of the “M” type patch and the bottom end of the “F” type patch, so the parameters L6, L5 having a large influence are subjected to parameter analysis. The other parameters are also like this. The effects of different values of L6, L5, L7, and Lg on the return loss of the antenna are shown in Figures 7–10. Below the graph is an analysis of the effect of different

values of the parameters. Different effects of different lengths are described in the paper. That’s all. Thank you.

16.  Conclusion – lacks rigour

Response: Conclusion has been revised. That’s all. Thank you.

Round 2

Reviewer 3 Report

1.       Language needs proof reading

2.       Table 1, Table 5 - #?

3.       Figure 2 – Rotate them for easy reading. Provide a key.

4.       Table 2 – format. Parameters – Why W and Wf? W3 and W4 – why h and a? Why missing values for W5 and W6? Very hard to read. Use vertical lines to separate the 3 sets of parameters and their corresponding units. Why L, Lf and Lg?  Discussion for Table 2 – insufficient.

5.       Figure 7 – not legible.

6.       Table 3 – provide a key  for the variables used.

7.       Page 12 – As can be seen …….. has a good radiation characteristic – elaborate and justify why it is good.

8.       5. Conclusion and Future Work – add some discussion of ways this work could be extended

9.       References – need more current resources

Author Response

1.     Language needs proof reading.

Response: The English language has been re-calibrated. That’s all. Thank you very much.

2.     Table 1, Table 5 - #?

Response: Table 1, Table 5 - # indicates information not mentioned in the reference. ‘#’ has been re-marked in the paper. That’s all. Thank you very much.

3.     Figure 2 – Rotate them for easy reading. Provide a key.

Response: Figure 2 has been rotated for easy to read. The individual parameter sizes are also relabeled. Figure 2 of the revised paper shows the revised part. That’s all. Thank you very much.

4.     Table 2 – format. Parameters – Why W and Wf? W3 and W4 – why h and a? Why missing values for W5 and W6? Very hard to read. Use vertical lines to separate the 3 sets of parameters and their corresponding units. Why L, Lf and Lg?  Discussion for Table 2 – insufficient.

Response: The values for W5 and W6 are not lost. Table 2 of the paper lists the detailed data values of all parameters. I have used vertical lines to separate the 3 sets of parameters and their corresponding units. W3, W4, L, Lf, etc. are an alphabetic representation of the basic dimensions of the antenna. First, design a structure that produces a three-band antenna. And the specific size was derived from High-Frequency Electromagnetic Simulation Software (HFSS). Then optimize the parameters by HFSS to find the parameters that best meet the design requirements. Further analysis of the discussion of Table 2, the revised version of the paper shows the part of the discussion. These specific values are the optimal solutions for software debugging. There may be no more explanation. That’s all. Thank you very much.

5.     Figure 7 – not legible.

Response: Figure 7 has been re-edited for easy to read. The revised picture can be found in Figure 7 of the revised paper. That’s all. Thank you very much.

6.      Table 3 – provide a key for the variables used.

Response: In Table 3, Ang and Mag represent the phase and amplitude at the resonance points in polar coordinates, respectively. Rx represents the input impedance of the antenna. Marked in the revised version of the paper. That’s all. Thank you very much.

7.     Page 12 – As can be seen …….. has a good radiation characteristic – elaborate and justify why it is good.

Response: A discussion of the advantages and disadvantages of radiation characteristics has been given in the revised paper. Please review the discussion of the revised paper.

8.       5. Conclusion and Future Work – add some discussion of ways this work could be extended.

Response: Conclusion and Future Work have been added and have some extensions to their future development. That’s all. Thank you very much.

9.     References – need more current resources

Response: References [16], [17] were published in 2018, and two additional references published in 2018 [21] and [22] were added. There are also several articles published in 2017 and 2016. That’s all. Thank you very much.
